# Automated Measurement of Geometric Features in Curvilinear Structures Exploiting Steger’s Algorithm

**DOI:** 10.3390/s23084023

**Published:** 2023-04-16

**Authors:** Nicola Giulietti, Paolo Chiariotti, Gian Marco Revel

**Affiliations:** 1Department of Mechanical Engineering, Politecnico di Milano, Via La Masa 1, 20156 Milan, Italy; 2Department of Industrial Engineering and Mathematical Science, Università Politecnica delle Marche, Via Brecce Bianche 12, 60131 Ancona, Italy

**Keywords:** vision-based measurement, geometric feature measurement, line width measurement, Steger algorithm

## Abstract

Accurately assessing the geometric features of curvilinear structures on images is of paramount importance in many vision-based measurement systems targeting technological fields such as quality control, defect analysis, biomedical, aerial, and satellite imaging. This paper aims at laying the basis for the development of fully automated vision-based measurement systems targeting the measurement of elements that can be treated as curvilinear structures in the resulting image, such as cracks in concrete elements. In particular, the goal is to overcome the limitation of exploiting the well-known Steger’s ridge detection algorithm in these applications because of the manual identification of the input parameters characterizing the algorithm, which are preventing its extensive use in the measurement field. This paper proposes an approach to make the selection phase of these input parameters fully automated. The metrological performance of the proposed approach is discussed. The method is demonstrated on both synthesized and experimental data.

## 1. Introduction

The line detection task is typically addressed in computer vision by two image filtering approaches: edge detection and ridge detection. Despite being similar in terms of the final goal, these two methods embed essential differences. An edge filter is typically a first derivative operator that measures how fast the intensity level changes across the entire image. These filters (e.g., Sobel, Canny, Roberts, etc. [1], to cite some) detect the boundaries between areas of different intensity values, e.g., high and low gray values, but their output consists of a double line, one on each side of the target. Contrarily, ridge detection algorithms work as second-derivative operators, and provide a single line as output for each target line, since they detect lines that are darker or brighter than their neighboring pixel. This feature has made ridge detectors very suitable not only for line detection tasks, but also for measurement purposes (e.g., line width, line extension, etc.), because it allows detection of the centerline position with high accuracy.

The computer vision literature has exposed several ridge detection algorithms over the years. These algorithms are also widely used in industrial contexts to automatically perform what used to be manual operations [2,3,4]. One well-established ridge detection algorithm that can locate, connect, and measure several geometric features of lines, e.g., width, is the well-known Steger’s line detection algorithm [5]. This algorithm, which makes the extraction of curvilinear structures possible, has found numerous applications in various fields since 1998. In [6], Zhang et al. use this method for detecting edges in images to develop a novel method for image resizing that preserves the image edge structures. This method is also widely used in medical imaging. Dobbe et al. [7] exploit the algorithm to detect the vessels’ centerline in the analysis of microcirculatory images. Fleming et al. [8] propose an application to the analysis of skin lesions, while Zhang [9] applied it to the quantitative measurement of neurites. Examples of microcirculatory geometry [7] and ophthalmic applications in the extraction of blood vessels [10] are also present. In the civil sector, the Steger’s line detector has been used for the identification of roads from satellite images and for the study of traffic with the automatic recognition of lines of cars [11]. In physics, the Steger’s algorithm has been used for recognizing gravitational waves from the study of time-frequency diagrams [12], and in document analysis to automate the process of digitizing lines in engineering drawings [13]. In machine vision, it has been highly exploited in 3D reconstruction techniques through stereo vision [14,15] and laser structured light techniques [16,17,18,19]. The Steger algorithm is used in [20] to delineate the central line of cracks and fractures in images using the concept of fractional differential. In [21], the line detection algorithm is compared with a deep neural network for the identification of filamentous complexes of proteins.

Despite its widely recognized benefit, the Steger’s algorithm also presents some drawbacks. Indeed, the algorithm requires accurate tweaking of its input parameters, which mainly relates to the width and contrast, with respect to the background, of the line to be identified in the image. The incorrect selection of these input parameters could not always guarantee the correct identification of the centerline or the correct identification of the width of the curvilinear structure. This has prevented the use of Steger’s approach in fully automated measurement systems targeted at the extraction of the geometrical features of structures with varying width and contrast levels.

In this context, this work aims at overcoming these issues by proposing a fully automated strategy providing the capability to select the optimal input parameters for the Steger algorithm, thus paving the way to its exploitation in measurement applications targeting variable-width/contrast curvilinear structures, such as, for example, cracks or scratches over surfaces, laser blade profile measurement, checking the alignment between surfaces, and so on. The developed method is restricted to the measurement of a single curved feature in an image. The paper is organized as follows: Section 2 discusses the theory behind the approach proposed. Section 3 discusses the metrological characterization of the proposed method, while Section 4 presents the results of the experimental campaign set up to validate the working strategy. Section 5 will draw the main conclusions of the work.

## 2. Optimizing Steger’s Algorithm Input Parameters

Lines in 2D images can be modeled through different characteristic profiles that involve the gray values along the direction perpendicular to the line. In [5,22], Steger discusses asymmetric bar-shaped, parabolic and Gaussian line profiles. In this paper we consider only bar-shaped profiles because they are more common in real applications and they effectively represent profiles identifiable on cracks, scratches or defects on surfaces, which turn out to be high-contrast lines with very sharp edges on the framed images. Equation (Equation 1) represents an asymmetrical bar-shaped line profile (dark line over a bright background) of half-width *w*, asymmetry a∈[0,1), and line contrast *h* with respect to the background. Note that the same considerations will also apply to bright lines on dark backgrounds (there is a minus sign in the Equation (Equation 1)) and that *w* represents the distance of the centerline to the side edges, so the actual line width will be 2w. The asymmetry parameter *a* represents the asymmetry of all the lines which have different surrounding intensity values; this parameter will be equal to zero if there is no asymmetry. Figure 1 shows an example of an asymmetrical bar-shaped line profile (dark line over a bright background) of half-width w=1 pixel, asymmetry a=0.2, and line contrast h=1 with respect to the background.
(1)$$f_{a}\left(x\right)=\left\{\begin{array}{cc} 0, & x<-w\\ -h, & |x|\leq w\\ -ha, & x>w \end{array}\right.$$

The original Steger’s line detection algorithm is very well-described in [5,22]. Apart from the target image, the algorithm obtains as input the contrast value *h* (defining the contrast of the line with respect to the background) and the σ parameter, which is the parameter defining the aperture of the Gaussian kernel to be convoluted with the original image. The algorithm provides as output the position and width of each line detected with sub-pixel accuracy. The contrast value depends solely on the intensity value difference between the line and the surrounding pixels. The contrast is related, in the Steger algorithm [5], to an upper threshold *u* and a low threshold *l* value, which represent the hysteresis threshold parameters. The low threshold value can be set equal to 10% of the upper threshold value and relate to the contrast according to Equations (Equation 2) and (Equation 3).
(2)$$u=h\left(\frac{2\sqrt{3}}{\sqrt{2\pi }\sigma^{2}}e^{-3/2}\right)$$
(3)l=0.1u

Any pixel whose second derivative is above the upper threshold will be marked as a pixel belonging to the line, as well as any other pixel whose intensity value lies between the higher and the lower thresholds, but is close enough, i.e., it is connected, to a pixel whose intensity value is above the higher threshold. All the other pixels of the image are not considered as pixels belonging to a line. The relations between *u*, *l* and the line contrast *h* suggest that, if the contrast of a line with respect to the background is high, it will be easier for the algorithm to isolate the line from the surrounding background.

If the line contrast can be easily estimated, e.g., by calculating the histogram associated with the intensity distribution of the image, the σ value strongly depends on the width of the line to be detected. This latter value can not be known a priori. Choosing the σ parameter involves the definition of a σ upper threshold. In fact, a σ value too high will cut out any line whose width is below a certain width, since the smoothing is too severe. Moreover, Steger establishes a lower threshold on the σ value through the following inequality (Equation 4), which represents the point at which the convolution function between the image and the second derivative of the Gaussian kernel reaches its maximum.
(4)$$\sigma \geq \frac{w}{\sqrt{3}}$$

The σ parameter therefore assumes the role of a scale-space parameter that drives the width range of the targeted line.

### 2.1. Automating a Line Width Measurement

A dedicated analysis was performed to test the Steger’s line detector dependency on the σ value. The rationale of the analysis is reported in the flow chart shown in Figure 2. Since the contrast value of the synthesized line is known, it is possible to separate the contributions of the various input parameters of the algorithm.

A synthesized line is generated with known width and asymmetry. The line contrast *h* is known, so the upper and lower threshold parameters can be calculated analytically for each σ value according to the Equations (Equation 2) and (Equation 3). Starting from $\sigma_{i}=\sigma_{1}=\sigma_{min}$, the Steger’s line detector algorithm is applied to the image. The mean width $\bar{w}\left(\sigma_{i}\right)$ of the detected line is then stored. The mean width is calculated as the arithmetic mean of all widths calculated by the Steger algorithm at each of the *m* line points identified.
(5)$$\bar{w}\left(\sigma_{i}\right)=\frac{1}{m}\sum_{j=0}^{m}w_{j}\left(\sigma_{i}\right)$$

The σ parameter is then increased by an incremental value *s*. These steps are repeated for increasing $\sigma_{i}$ values up to $\sigma_{i}=\sigma_{max}$. The interval $\sigma_{min}$–$\sigma_{max}$ is chosen to be compatible with the line width under analysis. In this way, it is possible to obtain the intersection of the function $\bar{w}=\bar{w}\left(\sigma \right)$ with the function reported in expression (Equation 4) in order to obtain the ${\bar{w}}_{s}$ and $\sigma_{s}$ values that fulfill the Steger’s inequality. The approach described above was tested on a synthesized horizontal line with a constant width of 5 pixels, and assuming $\sigma_{min}=0.4$, $\sigma_{max}=5$, s=0.2, and a=0 (Figure 3). The $\bar{w}\left(\sigma \right)\neq 0$ values obtained were then linearly interpolated, making it possible to generate a $\bar{w}\left(\sigma \right)$ function. The intersection of Steger’s inequality (Equation 4) with the $\bar{w}\left(\sigma \right)$ function, returns a value of ${\bar{w}}_{s}=5.019$ pixels. This results in a line width estimation error of 0.019 pixels.

To verify the potential sensitivity to the incremental value *s*, this value was varied from 0.05 to 1 with step 0.05 for the same σ range. The results of this iterative analysis are reported in Figure 4 in terms of absolute difference between the measured value ${\bar{w}}_{s}$ and the target value. As shown in Figure 4, no matter the step *s*, the error remains very small and well below one pixel. As for the number of iterations, a value of s=0.05, with $\sigma_{min}=0.4$ and $\sigma_{max}=5$, results in 92 iterations, while it reduces to 4 iterations with s=0.1. If the very same procedure is to be applied in an actual line width measurement, a trade-off between the required accuracy and the execution time of the algorithm, which actually increases by an order of magnitude by reducing the iteration step, will necessarily be needed.

The test was then repeated to verify the sensitivity to the asymmetry parameter *a*. The following parameters were kept fixed, i.e., $\sigma_{min}=0.4$, $\sigma_{max}=5$, s=0.2, and $\bar{w}=5$ pixel, while the asymmetry parameter *a* ranged from 0 to 1. A step of 0.05 was selected as the increasing step of the *a* parameter. The results, in terms of absolute difference between measured value ${\bar{w}}_{s}$ and target value, are reported in Figure 5. As shown in the graph, the error increases as the imposed asymmetry increases.

Last but not least, the following parameter values were considered: the $\bar{w}$ parameter was made to vary from 2 to 20 pixels, adopting an incremental step of 1 pixel, with a=0, s=0.1, $\sigma_{min}=0.4$, and $\sigma_{max}=15$. As the graphs in Figure 6 shows, the absolute error $e_{s}$ increases as the target width increases. The $e_{s}$ is calculated according to the following equation:(6)$$e_{s}=\left|{\bar{w}}_{s}-\bar{w}\right|$$

Nevertheless, the percentage error remains well below 1% (see Figure 7). If we consider the percentage error, an average percentage value of 0.47% is identified for the range of width investigated.

### 2.2. Optimizing the Sigma Parameter Selection

Section 2.1 demonstrated that is possible to exploit the Steger’s line detection algorithm to calculate the position and width of curvilinear structures in 2D images automatically with no manual selection of input parameters. This section investigates the possibility to further optimize this process to improve the metrological performance of a measurement system exploiting the approach.

By intersecting the inequality (Equation 4) with the $\bar{w}\left(\sigma \right)$ function, which is obtained by performing a linear interpolation of $\bar{w}\left(\sigma \right)$ for $\bar{w}\left(\sigma \right)\neq 0$, it is possible to obtain the mean width of the target curvilinear structure (${\bar{w}}_{s}$). Although this technique is very robust and returns a low error even when varying the target mean width and the asymmetry of the synthetic line generated, it is easy to see that the point identified (Figure 3) is not the one that minimizes the overall width error. In fact, from the graph shown in Figure 3, it can be seen that the algorithm, with low σ values, fails to detect the line, thus leading to a null ${\bar{w}}_{s}$. From a certain value of σ, which from now on we will call $\sigma^{*}$ (the minimum sigma at which the algorithm identifies the line and then returns a width value greater than zero), the algorithm identifies the line and therefore its mean width. From here on, the measured ${\bar{w}}_{s}\left(\sigma \right)$ will tend to increase as the σ value used increases. If we sketch the error trend from $\sigma^{*}$ onward, we can see that the minimum error is located at $\sigma^{*}$ and not at $\sigma_{s}$ (Figure 8). More specifically, at $\sigma_{s}$, the error is $e_{s}=0.019$ pixel, while at $\sigma^{*}$ the error is $e^{*}=5\times 10^{-7}$ pixels.

To test the performance of the algorithm at different line widths, the following iterative analysis was performed. The $\bar{w}$ values were varied from 2 to 20 pixels in steps of 1 pixel, with a=0, s=0.1, $\sigma_{min}=0.4$ and $\sigma_{max}=15$. As the graphs in Figure 9 shows, the absolute error increases as the target width increases. The same happens for the percentage error (Figure 10).

This high performance can only be achieved through the use of a very fine step *s* (see Figure 11). In practice, it is not possible for all applications to iterate (e.g., by using a brute force approach) the line detection algorithm for several iterations, as the computing time would increase excessively, especially with high resolution images.

Figure 12 shows the $\left(\sigma^{*},{\bar{w}}^{*}\right)$ progression for a synthesized line width ranging between 2 and 20 pixels. The σ parameter incremental step was set as s=0.1.

By performing a linear regression on these points we obtain a straight line of equation r(σ)=2.551σ+0.458, resulting in an MSE=0.014 pixels (mean square error) and a squared correlation coefficient $R^{2}=0.999$. This interpolation line represents an optimization line for the σ parameter.

Given the results shown, it is possible to understand that a procedure to identify the optimal sigma parameter would pave the way to an automated use of the Steger algorithm. This said, we propose one possible optimization approach to tackle this issue. Other algorithms can be exploited with the same purpose. The one addressed in the paper was selected to ease implementation also in embedded systems characterized by low computing power. To better describe the the sigma optimization procedure proposed, let us consider the flow chart reported in Figure 13. Starting from an input image containing a curvilinear structure (1), the Steger’s line detection algorithm is applied iteratively (2). In this phase all the input parameters (*h*,*s*) must be defined according to the scenario. Specifically, the step *s* is chosen according to the maximum computing time allowed. At each iteration, the value of sigma, which starts at 0.6 (i.e., the lowest value that can be used by the algorithm [5]), is increased by *s*. In the loop, the horizontal distance $d_{i}$ of the point $\left(\sigma_{i},\bar{w}\left(\sigma_{i}\right)\right)$ to the optimization line r(σ) is then calculated (Equation (Equation 7)).
(7)$$d_{i}=\left|\frac{\bar{w}\left(\sigma_{i}\right)-0.458}{2.551}-\sigma_{i}\right|$$

The loop continues until the following condition is fulfilled: $d_{i}>d_{i-1}\wedge \bar{w}\left(\sigma_{i-1}\right)>0$. Thanks to this process, there is no longer any need to define a minimum and maximum value for the σ parameter in the iteration (2). The output of the process is $\sigma^{*}=\sigma_{i-1}$, i.e., the first sigma value at which the vector comes close to the optimization line and results in a mean width greater than zero. If the value of *s* chosen in (2) is sufficiently small, the value of ${\bar{w}}_{opt}$, which in this case coincides with ${\bar{w}}^{*}$, is obtained. If the value of *s* is high because the computing time is to be kept low, then, starting from the point $\left(\sigma^{*},{\bar{w}}^{*}\right)$ (Figure 14b) a horizontal line is drawn up to the optimization line (Figure 14c). The σ value corresponding to the intersection of this horizontal line with the optimization line represents the $\sigma_{opt}$ value (Figure 14d) (3). Once this value is obtained, the Steger’s algorithm can be applied to the input image with the optimal sigma value, obtaining the ${\bar{w}}_{opt}$ value (4) as output.

## 3. Metrological Performance Assessment

### 3.1. Target-to-Camera Relative Orientation Dependence

Changes in the target-to-camera relative pose give rise to increased errors in geometric features assessment. These errors, which are mainly due to perspective distortion issues, can be mitigated by applying perspective distortion mitigation approaches. However, these approaches typically require featured images to work properly. This condition is borderline when targeting images presenting only curvilinear structures. Given this, it is the authors’ opinion that it is worth discussing the target-to-camera dependence with no perspective distortion mitigation pre-processing, as this better represents the majority of the working conditions one might have when addressing the topic of measuring geometrical features on curvilinear structures. The dependence of the developed method to the target-to-camera relative orientation was determined through the measurement of a real curvilinear structure in a controlled environment. To this end, a 6-dof (degrees of freedom) anthropomorphic robot was used. A plate embedding a curvilinear feature of constant width (1.25 mm) was attached to the end-effector of the robot, while three fiducial markers were placed on the plate to provide pixel-to-millimeter conversion factors, as well as to make eventual perspective correction possible.

The reference system of the end-effector is shown in Figure 15. The *z* axis is the axis normal to the plane hosting the line. The origin of the axes reference is at the center of the line. The same optical system exploited in Section 4 was used as the imaging system. The reflex camera was placed on a tripod at a known distance from the target. The parallelism between the camera sensor surface and the target surface was guaranteed through a custom-made alignment system composed of a laser projector and a mirror (Figure 16). A laser was rigidly mounted on the target surface (3) and a mirror (2) was attached to the front surface of the imaging lens. Through manual control of the robot, the end-effector was positioned in such a way that the laser beam, reflecting on the mirror surface, scattered back to the emission point. After the alignment operation, the laser and mirror were removed and the acquisition started. To ensure repeatability on the mounting of the laser on the camera, a mounting tool was used. The tool made it possible to rigidly fix the laser on the target surface.

Three types of acquisitions were performed:Controlled rotation of the target surface from $-30^{\circ }$ to $+30^{\circ }$, with steps of $0.5^{\circ }$, rotating around the *x* axis, for a total of 60 images acquired;Controlled rotation of the target surface from $-30^{\circ }$ to $+30^{\circ }$, with steps of $0.5^{\circ }$, rotating around the *y* axis, for a total of 60 images acquired;Controlled rotation of the target surface with random rotation angles from $-30^{\circ }$ to $+30^{\circ }$ around the *x* and *y* axes, for a total of 360 images acquired.

The first two tests were carried out to separate the single contribution due to the relative angle laying between the surfaces. The third type of test, on the other hand, was used to estimate the type A uncertainty of the system as the target-to-camera relative pose angles vary. All these measurements were repeated at different distances: 700 mm, 1000 mm, 1300 mm, and 2400 mm. The approach discussed in the previous section was then applied to all the images acquired.

#### 3.1.1. Target Rotated around the *x* Axis

A controlled rotation around the *x* axis was imposed to the target surface. The rotation angles ranged from $-30^{\circ }$ to $+30^{\circ }$, with an angular step of $0.5^{\circ }$. The measured mean width value and the pixel-to-millimeter conversion factor were calculated on each image acquired for each rotation angle and for each working distance tested.

Figure 17 shows the dependence of the absolute error, i.e., the absolute difference between the mean width measured by the algorithm and the target width (1.25 mm), to the rotation angle around the *x* axis for each working distance tested. The outliers relating to the measurement carried out at a distance of 2400 mm are due to the fact that at this distance the algorithm is not always able to correctly identify the markers used. It is easy to notice that the measurement results do not depend on the target-to-camera relative distance. This is an intrinsic benefit of the ridge detection algorithm used. In fact, once the line position is detected, the algorithm measures the line width perpendicularly to its center through the identification of the line edges-to-background intensity transitions. This reduces the importance of the line width in terms of pixel numbers. Indeed, this concept is clearly highlighted in Figure 18, where the width of the line reported in pixels is shown for each rotation angle around the *x* axis and for each working distance tested. As it is logical to expect, when moving away from the target, the number of pixels identifying the line width decreases as well as the absolute error in pixels. This might be due to an incorrect parallelism between the measurement and the plane embedding the target line.

#### 3.1.2. Target Rotated around *y* Axis

A controlled rotation around the *y* axis of the reference system shown in Figure 15 was imposed on the target surface in order to cover an angular range from $-30^{\circ }$ to $+30^{\circ }$, with an angular step of $0.5^{\circ }$. The test was repeated for each working distance identified in the introduction of Section 3.1.

Figure 19 and Figure 20 show the width absolute error dependency, in mm and pixels, respectively, to the rotation angle around the *y* axis. Again, the absolute error was calculated as the difference between the mean width measured by the proposed algorithm and the target width (1.25 mm). The outliers relating to the measurement carried out at a distance of 2400 mm are due to the fact that at this distance the algorithm is not always able to correctly identify the markers used. The same considerations of Section 3.1.1 hold.

#### 3.1.3. Target Rotated around *x* and *y* Axis

To understand the effect of a combined rotation of the target around the *x* and *y* axes, a dedicated test was carried out. Pure random rotations around these axes were imposed on the target. The pure rotation movement of the end-effector was imposed to keep the same target-to-camera distance during all the different target orientations. Each angular value of the two axes could range from $-30^{\circ }$ to $+30^{\circ }$. The random rotation values were assigned according to a uniform distribution for each rotation axis. The test was repeated for all the four distances (i.e., 700 mm, 1000 mm, 1300 mm and, 2400 mm). This resulted in a total of 360 images collected. The proposed algorithm was then applied to each acquired image to calculate the average width of the detected line.

Figure 21 and Figure 22 show the trend of the standard deviation as the maximum random absolute angle varies. The standard deviation addressed at an angle $\theta_{xy,i}$ is estimated as the standard deviation of the distribution of the line widths identified in the ranges $-\theta_{xy,i}\leq \theta_{x}\leq \theta_{xy,i}\wedge -\theta_{xy,i}\leq \theta_{y}\leq \theta_{xy,i}$. The first figure shows the standard deviation in mm, the second in pixels.

Similarly to what happened with the fixed rotation at $\theta_{x}$ and $\theta_{y}$ angles, the target-to-camera distance does not affect the trend of the standard deviation of the dataset (Figure 21). Contrarily, the higher the rotation angle, the higher the standard deviation and the measurement error.

A further interesting aspect is linked to the analysis of the variability of the sigma value with respect to the target-to-camera relative distance. Figure 23 shows the dispersion of the values at each distance, normalized to the equivalent pixel width of the line. For example, if we consider a line that is 7 pixels thick, the optimal sigma value selected by the proposed algorithm varies from 1.0 to 2.4 (no matter the target-to-camera angle). The variability range spans from 5.8 to 8.4 for an equivalent line width of 30 pixels (i.e., image taken from a closer distance). In this latter case, the sigma value can assume 13 different values, if we assume s=0.2. This clearly demonstrates how fundamental the choice of a correct sigma value is and how much this can vary a lot even in images taken at the same distance from the target. The Steger algorithm, without the proposed automatic optimization technique, could never have returned the results obtained without knowing a-priori the optimal sigma value.

### 3.2. Performance Comparison on a Reference Target

The metrological performance of the developed method was tested on a reference target by performing multiple acquisitions at different target-to-camera distances while keeping null the target-to-camera $\theta_{x}$ and $\theta_{y}$ angles. A reference target was manufactured in fused deposition modeling (FDM) 3D printing. As shown in Figure 24, the target embedded a central groove of 5 mm (nominal design value) in width.

The reference target and the camera were mounted on an optical bench, and their relative distance was varied between 700 mm and 1675 mm, at discrete steps of 25 mm. An image was acquired at each target-to-camera distance, thus resulting in a total dataset of 40 images (i.e., first image acquired at 700 mm, last image acquired at 1675 mm). This dataset made it possible to estimate the expanded uncertainty U=0.018 mm (coverage factor k=2) associated with the measurement system. Moreover, the sample mean $x_{i}=4.657$ mm of the groove width distribution was also estimated. Given the difference with respect to the nominal width of the groove (5 mm), and being aware of the uncertainty associated to the dimensional features of the FDM 3D-printed parts, two further high-precision optical-based measurement systems were also exploited to assess the reference groove width value:A custom Telecentric-based imaging system (TIS—Figure 25) in backlight arrangement (declared expanded uncertainty: $U_{TIS}=25.5$ μm), used for “in-production” dimensional quality control [23];A Wenglor MLSL132 laser profilometer (LP—Figure 26—measured expanded uncertainty at working distance of 150 mm: $U_{LP}=11$
μm).

Table 1 reports the width values measured by the three systems. It is worth saying that all the width values of Table 1 are width values averaged over several cross-sections at different heights along the groove. Since both TIS and LP are affected by the target-to-device pose, a double testing approach was adopted: (a) The same alignment procedure described in Section 3.1 was exploited for guaranteeing the target to be perpendicular to the optical axis of the camera of TIS. (b) The horizontal grooves were used as targets to align the laser line of LP on the target.

It interesting to notice that all the three devices shift to lower width values with respect to the nominal one (5 mm). Moreover, all the three measurements are compatible, thus demonstrating the validity of the approach developed.

This is even more evident if looking at Figure 27, which reports the averaged width values ($\hat{w}$) estimated using the proposed approach for different target-to-camera working distances. It is indeed well evident that the proposed approach makes it possible to obtain results compatible with those provided by the other instruments even at working distances varying over quite a wide range. To provide a clearer idea of the impact of the target-to-camera working distance on the acquired image we reported an example of such in Figure 28, where two images acquired at a target-to-camera working distance of 700 mm and 1675 mm are shown. Despite the clear size difference between the sub-areas of the images embedding the groove, the width values estimated are still compatible with those measured by TIS and LP.

## 4. Experimental Validation

The whole approach developed was tested on a test setup (Figure 29) specifically arranged to ensure the possibility to acquire images of grooves of variable width. Indeed, the target groove was artificially created by mounting two plastic parts (Figure 29 (3)) on the mobile and fixed components of a micrometric stage (Figure 29 (4)) (Newport 3-axis motion controller ESP300; uncertainty ±0.01 mm). The scene was framed with a camera (Figure 29 (1)) mounted on a tripod (Figure 29 (2)). The target was illuminated homogeneously, to avoid illumination gradient problems. The distance between the camera and the target was fixed to 500 mm. The camera used in this setup was a 24 Mp Nikon D7200 equipped with a 60 mm 2.8 Nikon Nikko Macro Lens.

Two fiducial ArUco [24] markers (5) were used to identify the target-to-camera relative pose and pixel-to-millimeter conversion factor. The width of the framed groove was varied through the micrometric stage between 0.1 and 2.5 mm. A step of 0.2 mm was adopted in varying the groove width, thus resulting in a total of 13 acquired images. The histogram of the region surrounding the line (corresponding to the real groove) was calculated to estimate the contrast value parameter *h*, as this is not known a priori. Figure 30 reports an example of the histogram (b) extracted for the region of interest containing the groove (3) in a image of the dataset (a). As the groove is darker with respect to the background, it is easy to identify the peak on the histogram representing those pixels belonging to the groove (1) and the peak referring to background pixels (2). The absolute difference between the grayscale values of the two peaks identified represents the *h* parameter to be used in the Steger algorithm.

Table 2 shows the results of the measurement campaign. The absolute error, i.e., the difference between the value set on the micrometric stage ($w^{\prime }$) and the value measured by the proposed method ($\bar{w^{\prime }}$), is given in mm ($e^{*,p}$) and in pixels ($e^{*,p}$). The pixel-to-millimeter conversion was performed using the conversion factor found during the calibration phase of the system. It seems there is no relationship between error and line width. The error is always well below one pixel. The mean error is 0.015 mm, equivalent to 0.383 pixels. Indeed, if comparing the groove width generated by moving the micrometric stage and its width value measured by the proposed method, a correlation coefficient $R^{2}=0.99$ is obtained, indicating a very strong correlation between the two measurements.

It is interesting to notice that the line detection algorithm starts to correctly identify the groove’s centerline from a certain σ value onward. For example, as shown in Figure 31, for σ=1.4, the algorithm does not identify the groove’s centerline, whereas for σ=1.5 it identifies the groove’s centerline throughout its whole length. In this case, $\sigma^{*}=1.5$ is shown to be the optimum σ value. As σ increases, the algorithm will continue to correctly identify the position of the groove’s centerline but will tend to overestimate its width.

To provide further validity to the approach proposed, an application to a real crack, i.e., a crack on a concrete surface (Figure 32) is discussed hereafter. This application is actually the target application for which the whole approach was developed, as the ultimate goal is the development of a measurement system specifically targeted to the assessment of the geometric features of superficial lesions, such as cracks.

A total of 100 pictures was acquired by varying the target-to-camera relative pose to verify the robustness of the approach to the use by a human operator. This pose variation was obtained by moving the camera closer and further away from the wall and taking the pictures from different heights. All pictures were taken with the same optical set-up used for the artificial groove (Nikon D7200 equipped with a 60 mm 2.8 Nikon Nikkor Macro Lens), but having a human operator handling the camera. The crack width distribution, normalized to the mean value is reported in Figure 33. The type A uncertainty associated with the measurements is estimated to be 0.0019 mm. If a coverage factor of k=2 is considered, an expanded uncertainty value of 0.0038 mm is identified.

## 5. Discussion and Conclusions

The aim of this work was to lay the basis for the development of a measurement system targeting an automated measurement of the geometric features of curvilinear structures such as those identifiable in building elements (e.g., concrete cracks). As the main drawback in exploiting the Steger algorithm (a well-known ridge detector approach providing line identification and width measurement) is that it is necessary to manually tune the parameters governing the algorithm, this paper has proposed an approach to overcome this issue, thus paving the way to a wider application of the algorithm for measurement purposes. Indeed, authors have proposed a method to automatically identify the two parameters that mostly affect a correct identification of the line and the assessment of its width, i.e., the *h* and the σ parameters. It was shown that the *h* parameter can be extracted by analyzing the histogram of the acquired image as the absolute difference between the two peaks on the histogram representing the curvilinear structure and the background. As for the σ parameter, an approach demonstrating how to identify the optimal parameter was developed and discussed in detail.

The metrological characterization of the proposed system was performed through dedicated tests (Section 3). A preliminary analysis on the influence of the target-to-camera relative pose was performed at first. A 6-dof anthropomorphic robot was used to acquire images at different relative target-to-camera angles and distances. This analysis of the datasets acquired made it possible to demonstrate the following:Contrary to what one might expect, the error in calculating the line width does not depend on the distance, and therefore on the number of pixels representing the line in the framed image. In fact, the width is not calculated through the convolution of the Gaussian profile, but through the use of an asymmetrical bar-shaped profile. The width of the bar-shaped profile therefore does not influence the calculation of the line width itself.The error increases as the angular misalignment between the measuring plane and the sensor plane increases. This happens because the perspective error becomes more marked and the pixel-to-millimeter conversion factor becomes more inaccurate.By varying the target-to-camera relative working angle within the range $\pm 30^{\circ }$, the maximum error is always below 0.100 mm, no matter the working distance, and therefore the number of pixels representing the line in the framed image.The standard deviation of the measurement increases in a quadratic manner as the angle of misalignment increases. With a maximum camera-target absolute angle of misalignment of $\pm 5^{\circ }$, a standard deviation of 0.003 mm is obtained, while with $\pm 30^{\circ }$, a standard deviation of 0.03 mm is estimated. This shows how, according to the application and therefore to the required metrological specifications, it is possible to set different acceptability ranges in terms of admitted angular misalignment.Based on the line width in pixels, it is possible to establish specific ranges of the sigma value representing the boundaries for searching the optimal sigma values. The maximum admissible angular range is $\pm 30^{\circ }$.

A further test was made to compare the performance of the solution with other non-contact measurement systems, specifically, a telecentric imaging system and a laser profilometer. The test was performed at different target-to-camera distances to demonstrate the invariance of the approach with respect to this variable. The results obtained show the compatibility of the measurements performed with the different systems, hence the robustness of the approach developed.

The whole method was tested on synthesized lines with an asymmetrical bar-shaped profile and a width ranging between 2 and 20 pixels. A maximum error of $e^{*}=1.4\cdot 10^{-5}$ pixels, corresponding to $e^{*}=7\cdot 10^{-5}\%$ in calculating the average line width was identified. The methodology developed was also validated on real lines of variable width (from 0.1 to 2.5 mm), resulting in a mean absolute error of 0.015 mm, equivalent to 0.383 pixels, in the measurement of the average line width. As further validation, the approach was tested on a real crack of a concrete surface. The type A uncertainty associated with the measurements was estimated by taking 100 picture of the crack, thus obtaining an expanded uncertainty value of 0.0038 mm (coverage factor k=2). The results obtained are highly promising. Authors are currently working on an approach to isolate the crack in the image, thus making it possible to further optimize the contrast parameter *h* value in the case of a surface containing multiple elements of variable contrasts, such as aggregates in concrete.

## Figures and Tables

**Figure 1 sensors-23-04023-f001:**
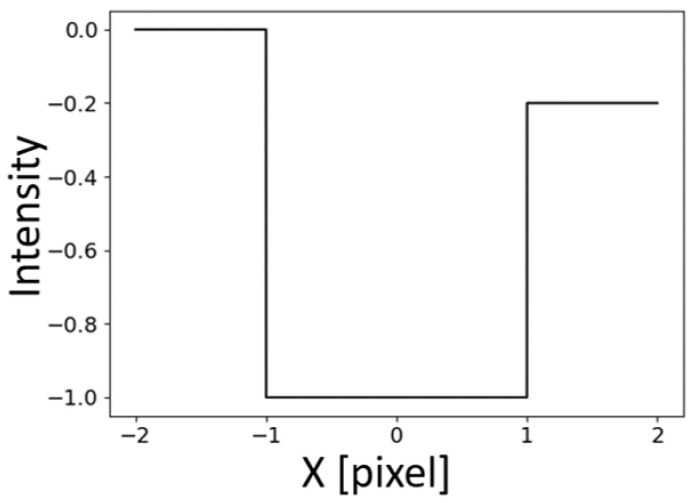
Example of an asymmetrical bar-shaped line profile (dark line over a bright background) of half-width w=1 pixel, asymmetry a=0.2 and line contrast h=1 with respect to the background.

**Figure 2 sensors-23-04023-f002:**
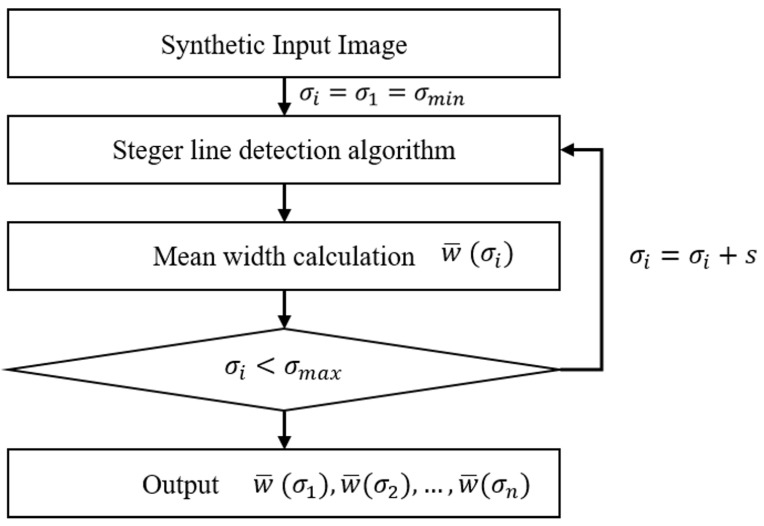
Procedure to identify the relationship between the line width $\hat{w}\left(\sigma_{i}\right)$ and the σ parameter.

**Figure 3 sensors-23-04023-f003:**
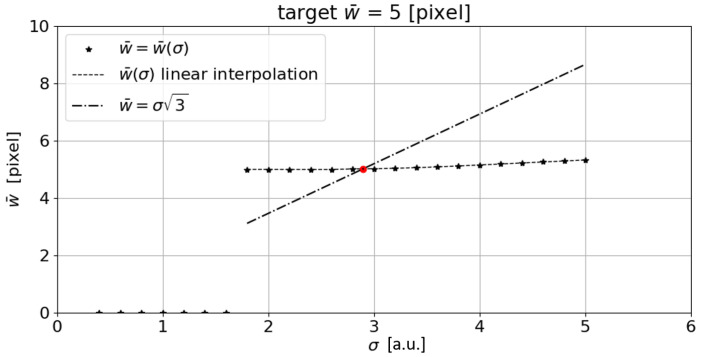
The measured $\bar{w}\left(\sigma \right)\neq 0$ are linearly interpolated, obtaining the $\bar{w}\left(\sigma \right)$ function (dashed line). Intersecting Steger’s inequality (Equation (Equation 4)) with the $\bar{w}\left(\sigma \right)$ function, returns ${\bar{w}}_{s}=5.019$ pixels (corresponding to the red dot). (a.u.: arbitrary unit).

**Figure 4 sensors-23-04023-f004:**
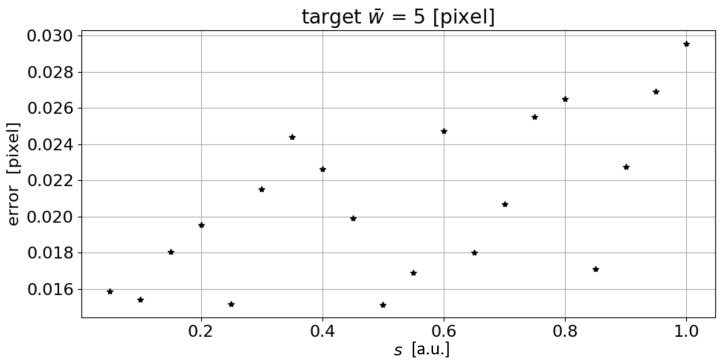
Variation of the absolute error in the measurement of the line mean width as the step *s* changes for the autonomous measurement of $\bar{w}$, according to the procedure described in Figure 2. (a.u.: arbitrary unit).

**Figure 5 sensors-23-04023-f005:**
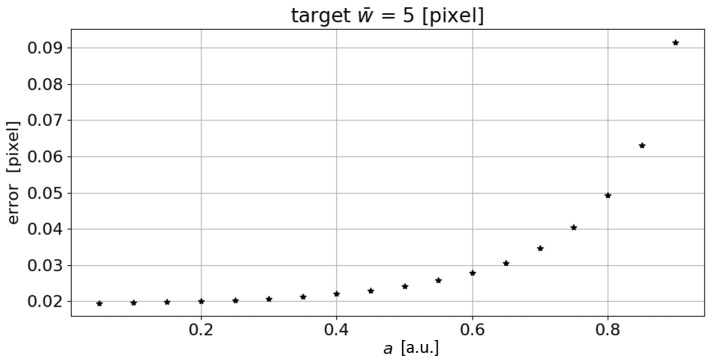
Absolute error trend in the measurement of the line mean width as the asymmetry *a* changes. (a.u.: arbitrary unit).

**Figure 6 sensors-23-04023-f006:**
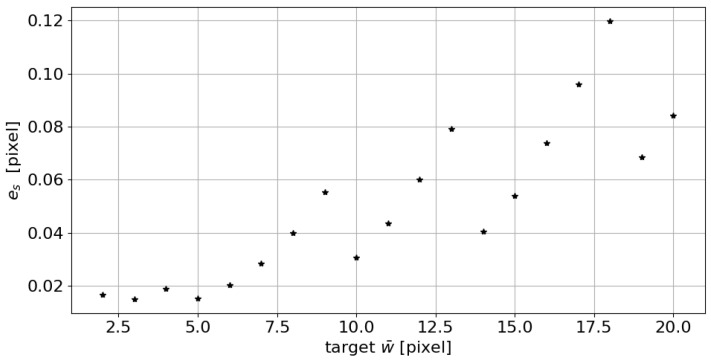
Absolute error $e_{s}$ trend in the measurement of the line mean width as the target $\bar{w}$ changes.

**Figure 7 sensors-23-04023-f007:**
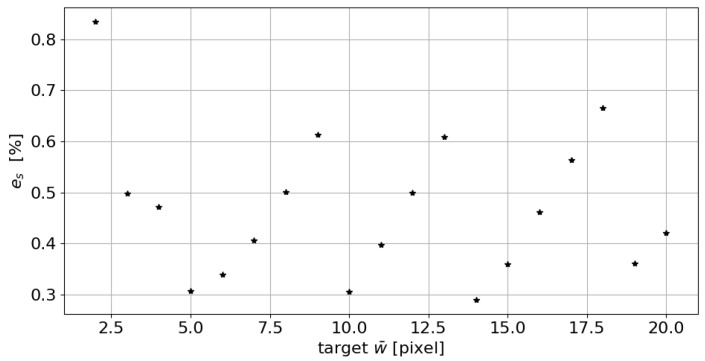
Absolute percentage error $e_{s}$ trend in the measurement of the line mean width as the target $\bar{w}$ changes.

**Figure 8 sensors-23-04023-f008:**
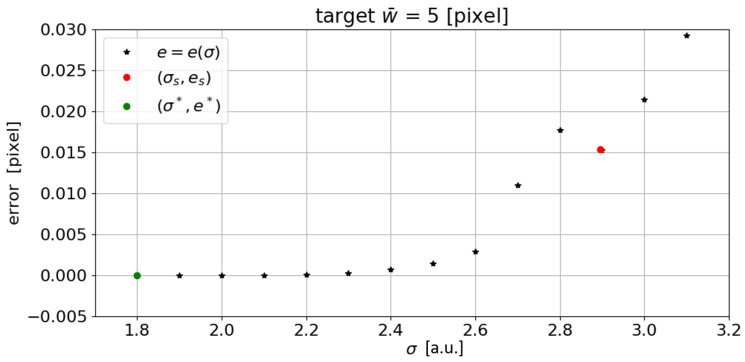
Detail of the graph in Figure 3, with the error shown in the abscissa. The minimum error is located at $\sigma^{*}$ (a.u.: arbitrary unit).

**Figure 9 sensors-23-04023-f009:**
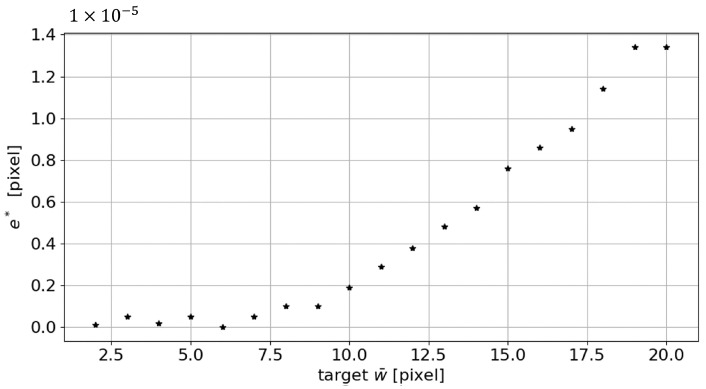
Absolute error $e^{*}$ trend in the measurement of the line mean width as the target $\bar{w}$ changes.

**Figure 10 sensors-23-04023-f010:**
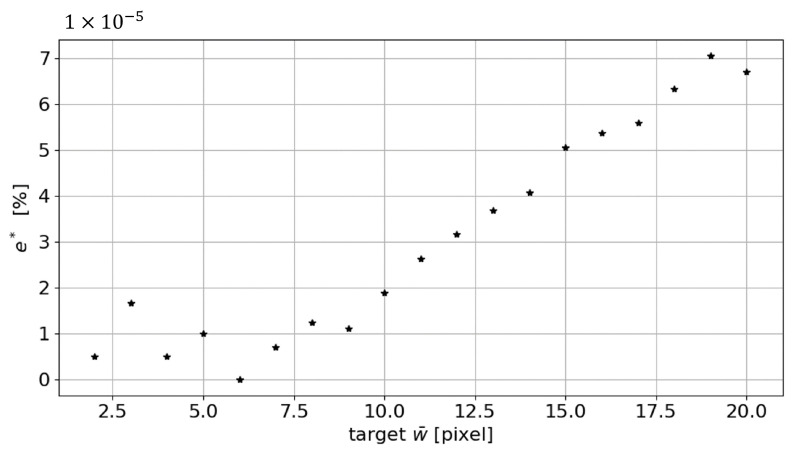
Absolute percentage error $e^{*}$ trend in the measurement of the line mean width as the target $\bar{w}$ changes.

**Figure 11 sensors-23-04023-f011:**
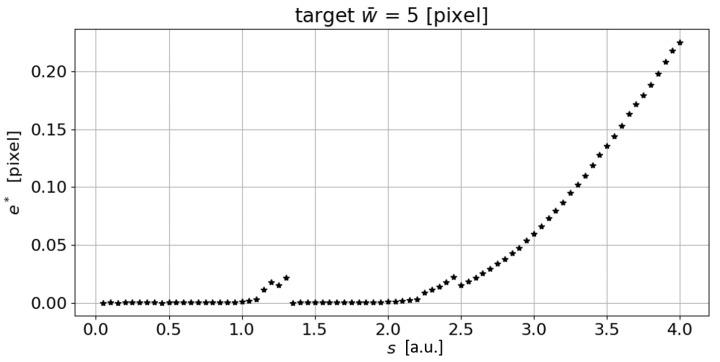
Error $e^{*}$ trend as the parameter *s* increases (a.u.: arbitrary unit).

**Figure 12 sensors-23-04023-f012:**
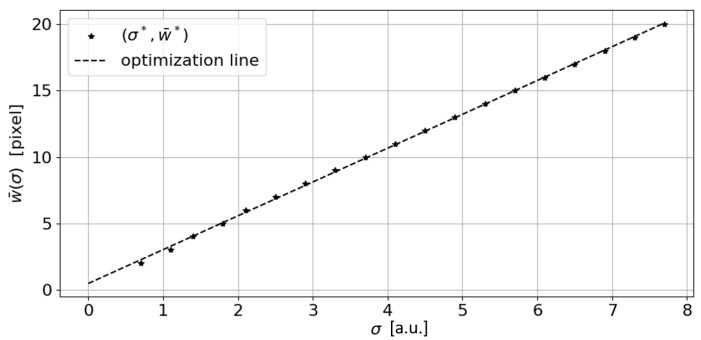
The σ optimization line (dashed line) vs. $\left(\sigma^{*},{\bar{w}}^{*}\right)$ points obtained by iteratively applying the algorithm on synthetic line widths ranging from 2 to 20 pixels with s=0.1 (a.u.: arbitrary unit).

**Figure 13 sensors-23-04023-f013:**
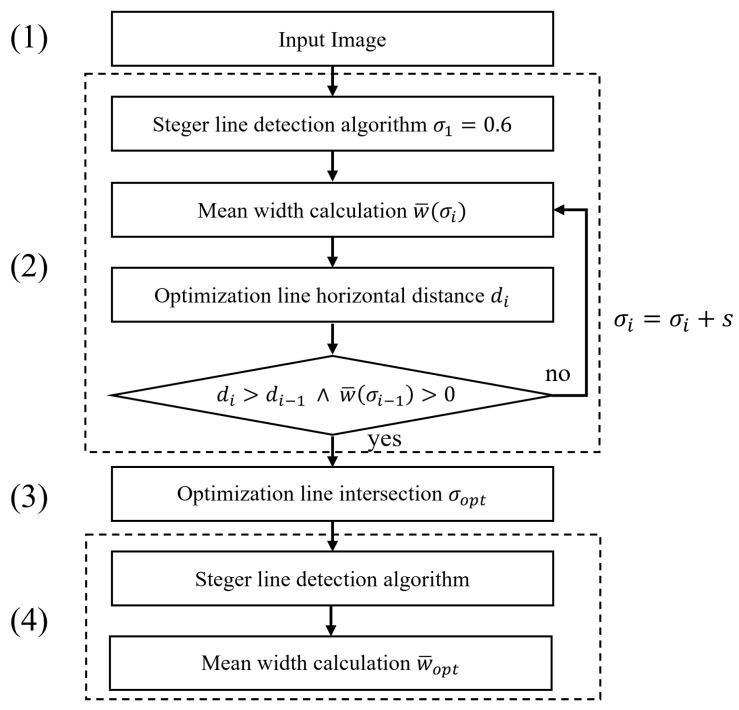
Proposed sigma optimization technique flowchart.

**Figure 14 sensors-23-04023-f014:**
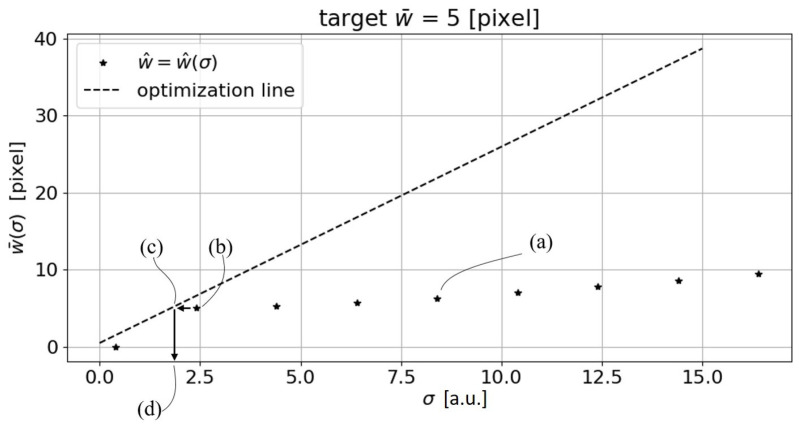
Example of sigma optimization technique application. (a.u.: arbitrary unit). After all necessary $\left(\sigma ,\bar{w}\right)$ (a) are calculated, starting from the point $\left(\sigma^{*},{\bar{w}}^{*}\right)$ (b), an horizontal line is drawn up to the optimization line (c). The σ value corresponding to the intersection of this horizontal line with the optimization line represents the $\sigma_{opt}$ value (d).

**Figure 15 sensors-23-04023-f015:**
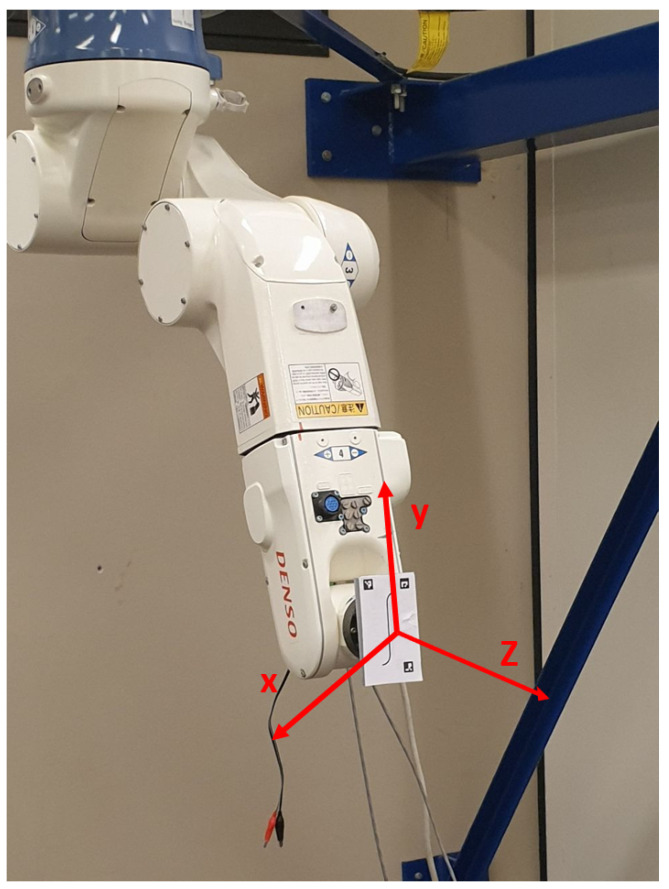
An anthropomorphic robot is used in order to define the working range of the developed device in a controlled environment.

**Figure 16 sensors-23-04023-f016:**
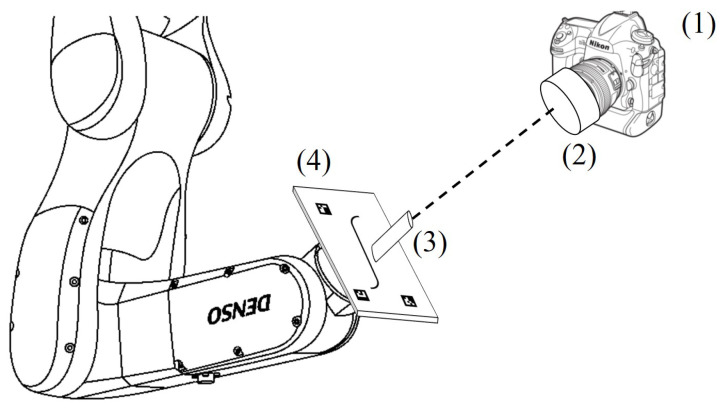
The image acquisition system (1) is placed at a known distance from the target (4). The parallelism between the sensor surface and the target surface is guaranteed through a system made of a laser (3) and a mirror (2). The two surfaces are to be considered parallel when the laser beam reflected by the mirror returns back to the source.

**Figure 17 sensors-23-04023-f017:**
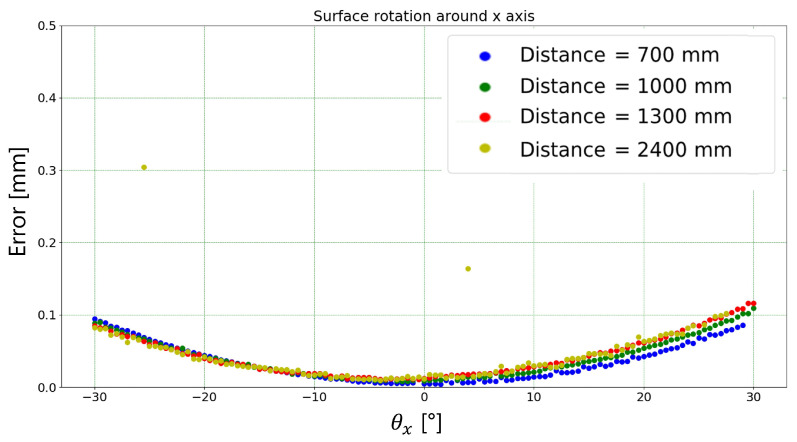
Absolute error (i.e., difference between mean width measured by the proposed algorithm and the target width) in millimeters for each rotation angle (around *x* axis) and each working distance tested.

**Figure 18 sensors-23-04023-f018:**
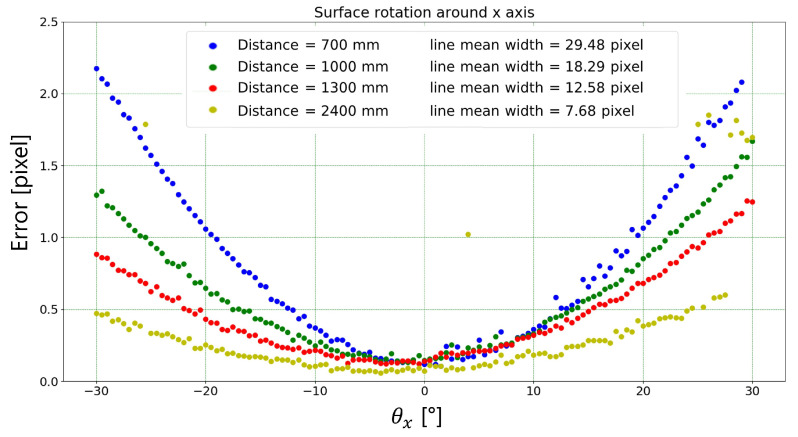
Absolute error (i.e., difference between mean width measured by the proposed algorithm and the target width) in pixels for each rotation angle (around *x* axis) and each working distance tested.

**Figure 19 sensors-23-04023-f019:**
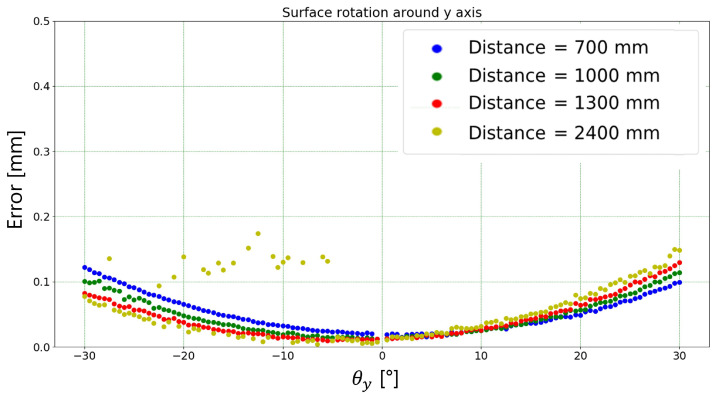
Absolute error (i.e., difference between mean width measured by the proposed algorithm and the target width) in mm for each rotation angle (around *y* axis) and each working distance tested.

**Figure 20 sensors-23-04023-f020:**
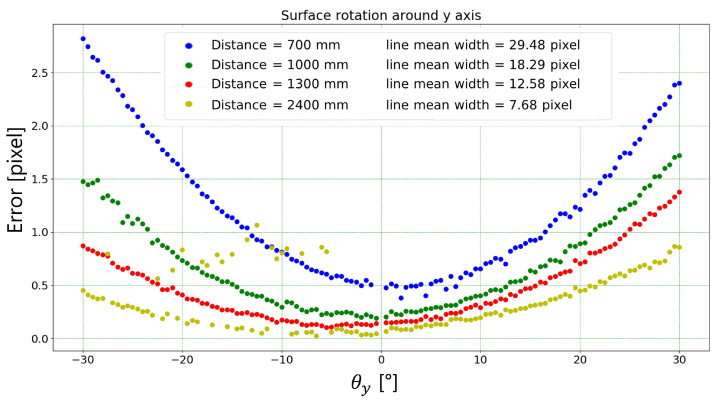
Absolute error (i.e., difference between mean width measured by the proposed algorithm and the target width) in pixels for each rotation angle (around *y* axis) and each working distance tested.

**Figure 21 sensors-23-04023-f021:**
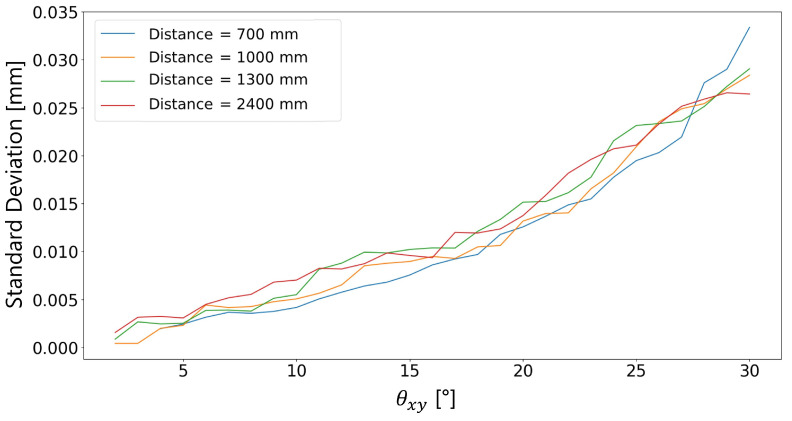
Standard deviation in mm vs. absolute angle.

**Figure 22 sensors-23-04023-f022:**
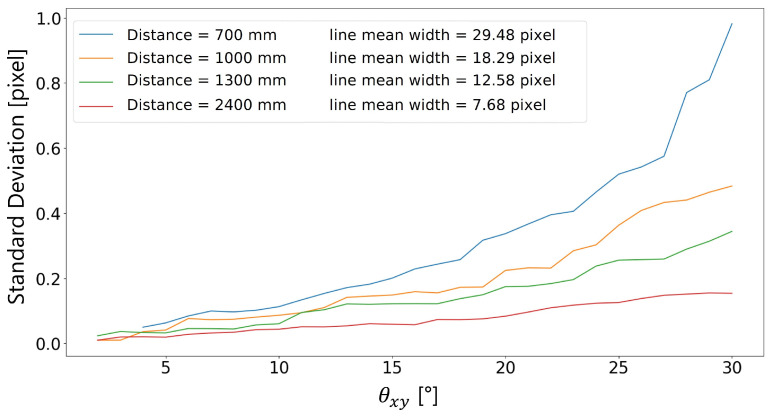
Standard deviation in pixels vs. absolute angle.

**Figure 23 sensors-23-04023-f023:**
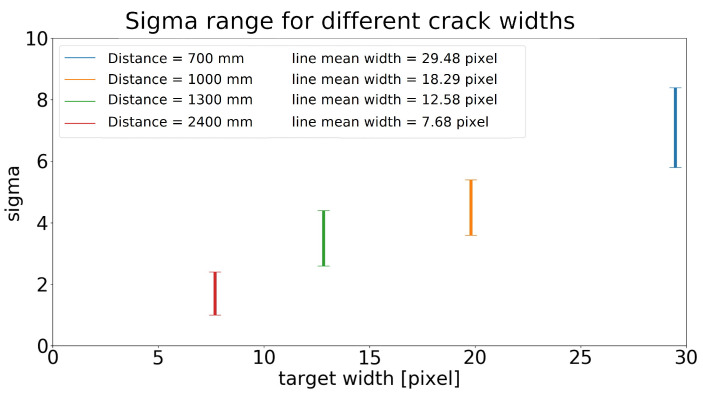
Dispersion of the values at each distance, normalized to the equivalent pixel width of the target line.

**Figure 24 sensors-23-04023-f024:**
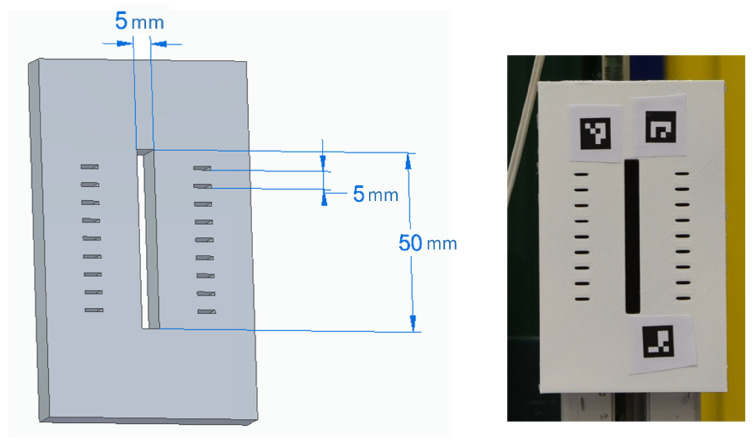
A reference target was produced in FDM 3D printing with a built-in central groove of 5 mm (nominal design value) width.

**Figure 25 sensors-23-04023-f025:**
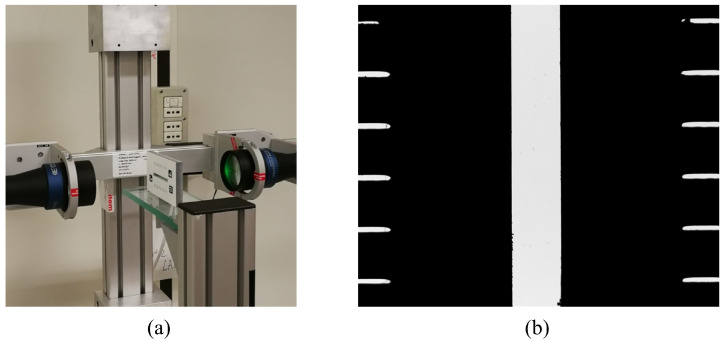
Telecentric-based imaging system: (**a**) optical set-up; (**b**) silhouette from backlight arrangement.

**Figure 26 sensors-23-04023-f026:**
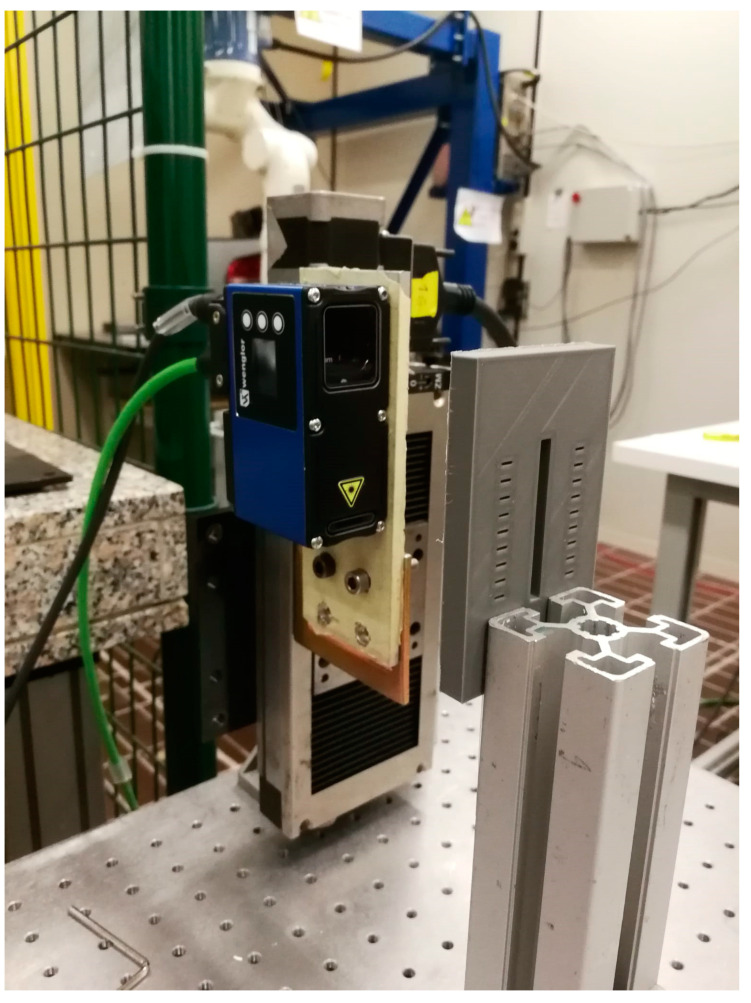
Test set-up of the Wenglor MLSL132 laser profilometer used to evaluate the value of the reference groove width.

**Figure 27 sensors-23-04023-f027:**
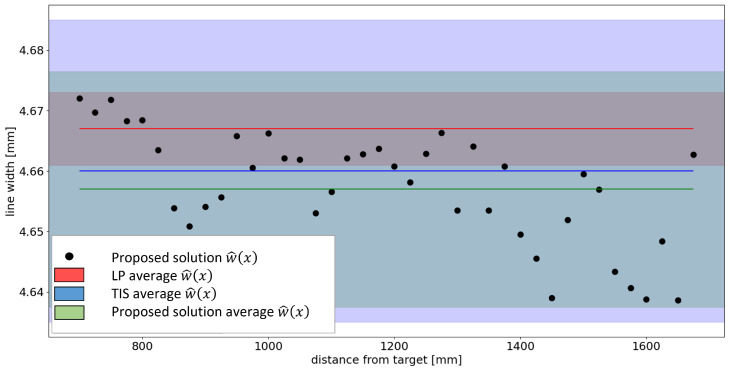
$\hat{w(x)}$ calculated using the proposed solution according to the distance *x* of the camera from the target. In the graph, the results of the measurements carried out are superimposed with the LP and TIS in terms of average value and uncertainty bounds.

**Figure 28 sensors-23-04023-f028:**
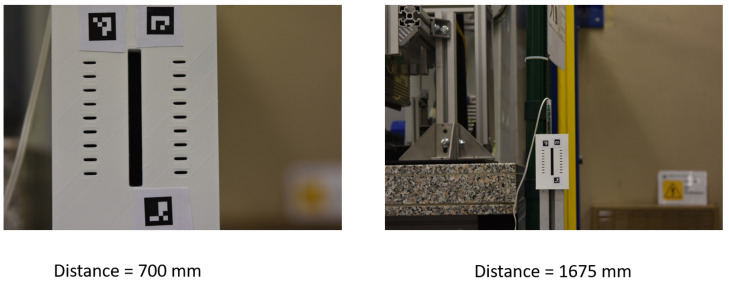
Acquisition setup. On the left, a picture framed at a distance of 700 mm. On the right a picture framed at a distance of 1675 mm.

**Figure 29 sensors-23-04023-f029:**
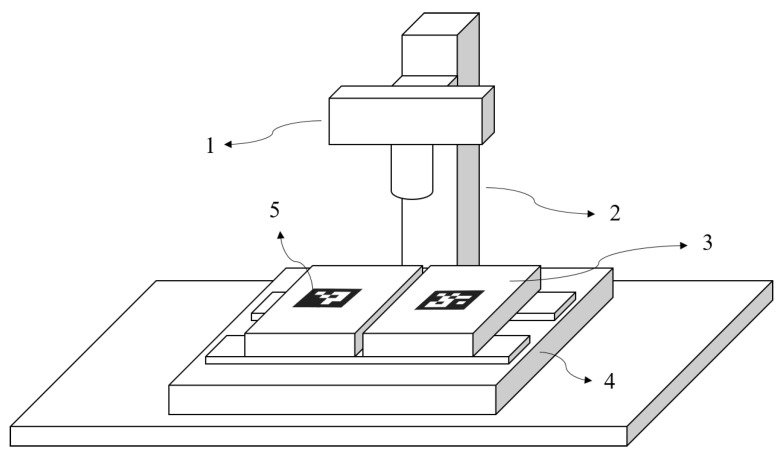
Experimental set-up to generate grooves of known, variable width. The target groove was artificially created by mounting two plastic parts (3) on the mobile and fixed components of a micrometric stage (4). The scene was framed with a camera (1) mounted on a tripod (2). Two fiducial ArUco markers (5) were used to identify the target-to-camera relative pose and pixel-to-millimeter conversion factor.

**Figure 30 sensors-23-04023-f030:**
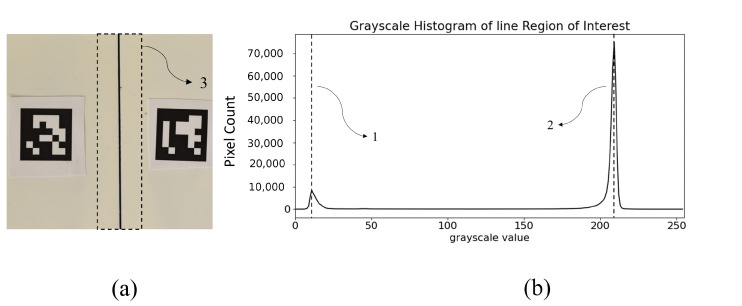
Histogram (**b**) of the grayscale image (**a**) of the area of interest, including the groove under test (3). Two main peaks can be identified in the histogram. On the left, the peak representing the area belonging to the groove (1); and on the right, the peak representing the area belonging to the background (2).

**Figure 31 sensors-23-04023-f031:**
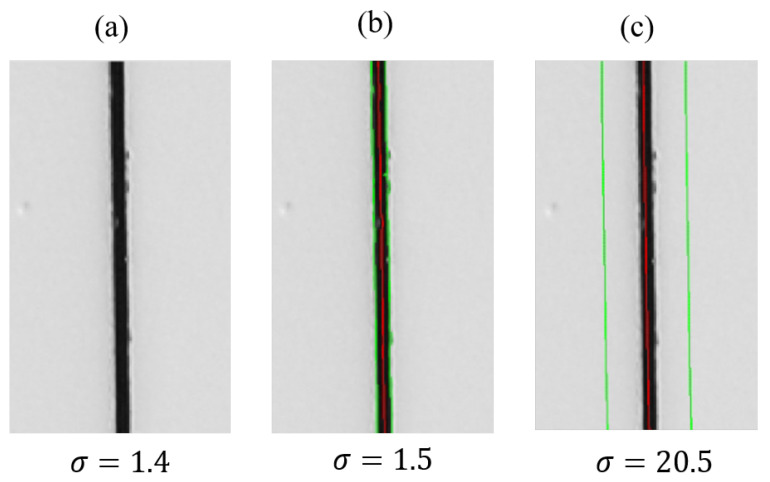
The optimization algorithm identifies the transition point between the unidentified groove condition (**a**) and the correctly identified one (**b**). As the σ value increases, the algorithm continues to identify the centerline well but overestimates its width value (**c**), represented by the green lines in the figure.

**Figure 32 sensors-23-04023-f032:**
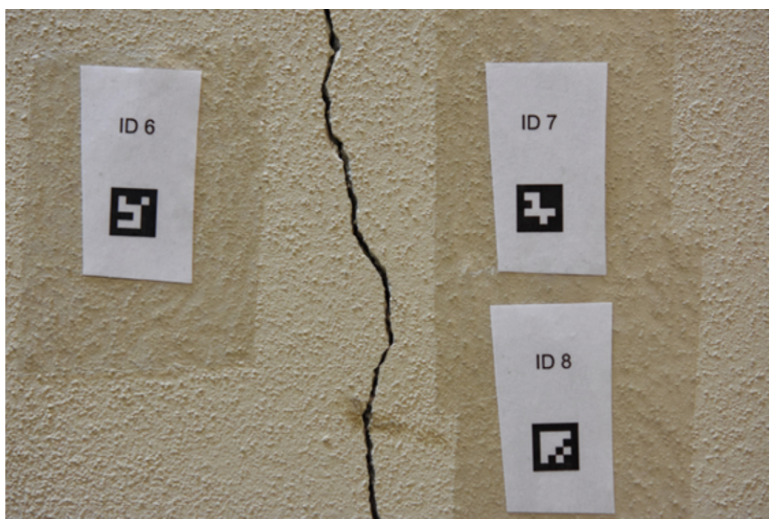
Target concrete wall crack.

**Figure 33 sensors-23-04023-f033:**
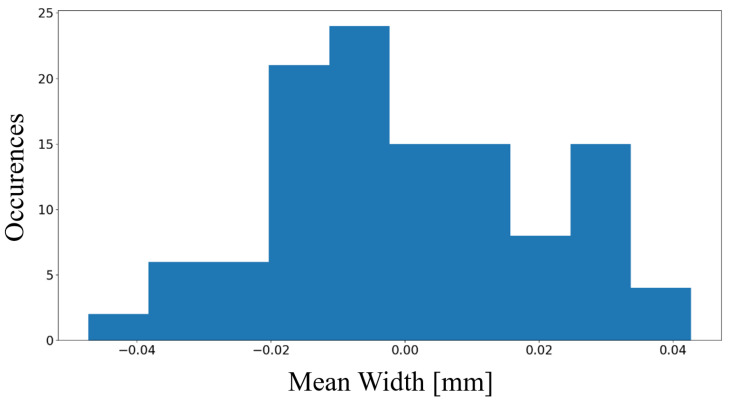
Distribution normalized with respect to the average value of mean width values measured on 100 images by the same operator.

**Table 1 sensors-23-04023-t001:** Comparisons between the average groove width measured with the three different devices.

	$\hat{w}$	$U_{\left(k=2\right)}$
	[mm]	[mm]
Proposed solution	4.657	0.018
TIS	4.660	0.025
LP	4.667	0.006

**Table 2 sensors-23-04023-t002:** Comparisons between the groove width set on the micrometric stage ($w^{\prime }$) and the corresponding groove width measured through the proposed vision-based method ($\bar{w^{\prime }}$). The absolute error is given in pixels ($e^{*,p}$) and mm ($e^{*}$).

$w^{\prime }$	$\bar{w^{\prime }}$	$e^{*}$	$e^{*,p}$
[mm]	[mm]	[mm]	[pixels]
0.100	0.128	0.028	0.726
0.300	0.301	0.001	0.032
0.500	0.473	0.027	0.699
0.700	0.674	0.026	0.686
0.900	0.872	0.028	0.741
1.100	1.091	0.009	0.229
1.300	1.294	0.006	0.154
1.500	1.506	0.006	0.164
1.700	1.697	0.003	0.088
1.900	1.918	0.018	0.480
2.100	2.096	0.004	0.100
2.300	2.280	0.020	0.534
2.500	2.513	0.013	0.346

## Data Availability

Not applicable.

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
