# Peer review of "Automated Measurement of Geometric Features in Curvilinear Structures Exploiting Steger’s Algorithm"

_sensors, 2023, doi:10.3390/s23084023_

Round 1
Reviewer 1 Report
In this paper, an automatic parameter optimization strategy of Steger’s algorithm is proposed for the measurement of geometric features in curvilinear structures. The optimal parameter combination is selected by analyzing the influence of various parameter changes on the measurement accuracy, and then the metrological performance of the system is assessed and its feasibility is verified by measurement tests. Finally, the concrete crack measurement experiment is carried out, and good results are obtained. However, the contents of 3.1.3 of this paper are suspected to be repeated with those of 3.1.1 and 3.1.2, which does not contribute to the further improvement of the overall framework of this paper. In addition, there are a lot of formatting and quoting errors. Here are the specific ones I found:
1. Is section 3.1.3 necessary? I think it overlaps with 3.1.1 and 3.1.2. Will the integrity of the paper be affected if it is deleted?
2. Periods are missing at the end of some figure legends (such as Figure 1, Figure 12, Figure 16 and Figure 17, etc.), and there is no corresponding picture in figure 27.
3. The format of some parts of the paper is not uniform. Whether there are Spaces between all numbers and units? Should unit and parameter names be in italics?
4. Whether "Figure 1" in lines 159 and 160 should be "Figure 2"?
5. Whether "mirror(4)" in Figure 15 should be "mirror(2)"?
6. How to calculate the "crack width" in Figure 17, 19, 23 and 24? I did not find the relevant description in this paper.
7. Does Figure 30-3 refer to the same thing as Figure 31-3? If so, the two refer to different places; If not, I did not find the relevant description of Figure 30-3 in this paper.
8. Some pictures are not clear enough, and the text in the pictures is too small (Figure 28). Besides, the size of the picture and table is not uniform, resulting in typesetting confusion.
9. Do the lasers and mirrors in Figure 15 need to be removed after alignment? Will they block the camera's view if you don't remove it? If removed, how to ensure that both are rigidly mounted each time?
Reviewer 2 Report
I would suggest to add a profile image to the equation 1 to visualize easily the profile.
1. What is the main question addressed by the research? The main question addressed in this paper is the automatic optimization of the two parameters for the Steger algorithm for the ridge extraction on images. This algorithm formulated 1998, tries to extract curvilinear features in images assuming three possible transversal profiles. Authors in this paper have worked on one of this types of profile the bar shaped which depends on two parameters h (line contrast) and Sigma which is the aperture of the gaussian filter used for the gradient computation of the image. The authors proposed an automatic selection procedure for these two parameters based on the width of the curvilinear feature to detect and on the contrast of this feature with respect to the background.
2. Do you consider the topic original or relevant in the field? Does it address a specific gap in the field? As far as I know there is no automatic procedure for the selection of these parameters to guarantee the extraction of the curvilinear feature. They also made some considerations regarding the influence of other aspects to the photographic capture which can influence the automatic detection of the line and the width estimation of the ridge, like target to camera distance and relative camera (sensor)-target angles. They call this "metrological characterization" and to my understanding is a good approach to prepare the method to be used in real practical applications. This part also guarantees a limited and known maximum error values expected in the real applications of the automatic procedure, depending of the angle of rotation between target and sensor planes.
3. What does it add to the subject area compared with other published material? As I said before, automatic selection of the parameters for Steger ridge detector, sensitivity analysis of the sigma (gaussian filter aperture) and h (line contrast with respect the background) is new to my knowledge. Other aspects like the synthetic simulations and lab measurements are useful for the planification of photographic capture for crack detection and maximum error estimation in advance.
4. What specific improvements should the authors consider regarding the methodology? What further controls should be considered? In my opinion the subject has been analyzed with enough deep and taking into account several configurations (relative distances and angles) and several line widths. Even the article at present, is a little bit longer than the average reader could need.
5. Are the conclusions consistent with the evidence and arguments presented and do they address the main question posed? Yes, the conclusion are supported by the very detailed synthetic tests and the laboratory ones, with specific numerical values for the metric parameters of the detected cracks with proposed method in different configurations.
6. Are the references appropriate? Yes they are. Curvilinear features are difficult to detect and characterize and is not a very visited subject in classic image processing field. In addition, newer techniques use to come from the Deep Learning environment but these techniques are based on big image databases with classified examples which are difficult to compile in special for real applications where and expert knowledge is required and multiple noise from other phenomena can appear. The automatic technique presented here with its metrological characterization, could serve to automatically label this kind of lines in the crack detection apps.
7. Please include any additional comments on the tables and figures. Fig 1 can include a graphic of the profile Eq. 2 and 3: I would suggest to change the equation 3 to simply I= 0.1 * u Line 71: There is a repetition of the dark word. Please correct.
Reviewer 3 Report
The article titled: "Automated measurement of geometric features in curvilinear structures exploiting Steger's algorithm" is interesting because it addresses an important scientific problem concerning measurement technology through image analysis. The subject is exciting from the point of view of automatic classification of defects occurring on the surfaces of various elements. The article discusses in detail the assumptions and measurement concepts adopted, which are then verified experimentally.
Specific comments:
Formulas appearing in lines 195, 194, 293, 300 should be written in a standard format such as formulas 1-4.
Since a lot of formulas are cited in the article, a list of designations would be useful.
In lines 244 245 the distance units are in cm in the paper generally uses mm so they could be standardized.
Figure 25 should be captioned in more detail.
In Figure 26, the markings (a) and (b) are disproportionately large, in addition, the drawings should be more accurately described in terms of their content. There are similar comments for Figures 27, 29 and 33.
In figure 30 there are markings from 1 to 5 no explanation in the caption.
Supplementation of the literature could be considered as there are few articles from the last 10 years.
